# Analysis of postoperative intraocular pathologies in patients with mature cataracts

**Min Seok Kim, Jung Hyoo Moon, Myung Won Lee, Kwan Hyuk Cho** [iD] *

Moon's Eye Clinic, Suwon, Gyeonggi-do, South Korea

* whmed@hanmail.net

## Abstract

### Purpose

To examine the prevalence and risk factors of intraocular pathologies after mature cataract surgery.

### Methods

The medical records of 115 patients (115 eyes) diagnosed with brunescent or white cataracts, who underwent surgery at a single primary center between January 2018 and August 2021 were retrospectively reviewed. Dense cataracts precluded preoperative fundus examination in all eyes; however, patients with fundus examination results within 3 months after cataract surgery were included. Logistic regression analyses were performed to identify factors associated with intraocular pathologies.

### Results

Intraocular pathologies were observed in 37 eyes (32.2%) 11.8 ± 13.9 days postoperatively. The most common abnormalities were drusen (6.1%), myopic degeneration (5.2%) and diabetic retinopathy (4.3%). Intraocular pathology in the fellow eye was associated with posterior segment pathology in mature cataract eyes (odds ratio, 47.72; $P < 0.001$).

### Conclusions

The prevalence of each intraocular pathology found after mature cataract surgery was unremarkable. This study provides clinically useful evidence for clinicians to explain the risk of posterior segment pathology in patients with mature cataracts.

## Introduction

A mature cataract, whether white or brunescent, is a totally opaque lens that causes severe visual loss. Due to increased surgical challenges and risks for complications, previous reports have focused on techniques for the safe surgery of mature cataracts [1, 2]. Besides surgical implications, mature cataracts also induce significant challenges during preoperative

**Data Availability Statement:** All relevant data are within the manuscript and its Supporting information files.

**Funding:** The authors received no specific funding for this work.

**Competing interests:** The authors have declared that no competing interests exist.

assessment, as severe media opacification impairs the view of the posterior segment of the eye. B-scan ultrasonography can be used as a preoperative diagnostic tool for posterior segment abnormalities; however, it is limited to rough intraocular lesions, such as posterior staphyloma, retinal detachment, and vitreous hemorrhage [3–7]. Preoperative assessment using visual electrophysiological techniques is also available; however, its accuracy and stability are affected by several factors, and the results are limited to the prediction of postoperative visual function [8–10]. This means that an accurate and detailed intraocular examination is only possible after cataract extraction in eyes with mature cataracts. However, the prevalence and risk factors of intraocular pathologies observed after mature cataract surgery have not been thoroughly evaluated. In this study, we reviewed the medical records of 115 patients with mature cataracts who underwent cataract surgery and evaluated the results of the postoperative fundus examinations.

## Materials and methods

This retrospective observational cohort study was approved by the Public Institutional Bioethics Committee, designated by the South Korea Ministry of Health and Welfare (No. P01-202109-21-019), and adhered to the tenets of the Declaration of Helsinki. The requirement for informed consent was waived due to the retrospective nature of the study.

We retrospectively reviewed the medical records of consecutive patients diagnosed with brunescent or white cataracts who underwent cataract surgery at the Moon's Eye Clinic between January 2018 and August 2021. Eligible criteria were the presence of a cataract too dense to allow visualization of the optic disc and retinal vessels on ultra-widefield fundus images (Optos PLC., Dunfermline, United Kingdom) or dilated fundus examination, image quality of macular optical coherence tomography (Spectralis OCT; Heidelberg Engineering, Heidelberg, Germany) 5 or less [11], and available fundus examination results within 3 months after surgery (S1 Fig).

Exclusion criteria were known posterior segment pathology, previous ocular surgery or penetrating trauma, and age of younger than 18 years. If both eyes were eligible, the eye with the later surgery was selected, because the fundus status of the fellow eye, which is one of the variables in this study, was available after preceding cataract surgery. Statistical analysis was performed using SPSS version 25.0 (IBM Corp., Armonk, New York, USA). Logistic regression analyses were performed to identify factors associated with intraocular pathologies. A univariate analysis was conducted for each variable, and a multivariate logistic regression analysis was conducted for factors with a $P$-value smaller than 0.20 in the univariate analysis. A $P$ value $< 0.05$ was considered to indicate significance.

## Results

A total of 11,054 patients underwent cataract surgery between January 2018 and August 2021 at the Moon's Eye Clinic. Among these cases, 115 cases (1.0%) with brunescent (n = 71, 61.7%) or white (n = 44, 38.3%) cataracts met the criteria for this study. All patients underwent cataract phacoemulsification and intraocular lens (IOL) insertion in the capsular bag, except for 9 (7.8%) patients who underwent additional vitrectomy and scleral fixation of the IOL (n = 4) or sulcus implantation of the IOL (n = 5) due to posterior capsular rupture. The baseline characteristics of the patients are shown in Table 1.

Intraocular pathologies were observed in 37 eyes (32.2%) 11.8 ± 13.9 days postoperatively, including 7 (6.1%) with drusen; 6 (5.2%) with myopic degeneration; 5 (4.3%) with diabetic retinopathy; 4 (3.5%) with glaucoma; 3 (2.6%) with epiretinal membrane and lattice degeneration, respectively; 2 (1.7%) with exudative age-related macular degeneration and retinal tear,

**Table 1. Baseline characteristics.**

| Characteristics | Value |
|---|---|
| Number of patients | 115 |
| Age (year) | 68.6 ± 12.8 |
| Gender (male/female) | 72 (62.6) / 43 (37.4) |
| Hypertension | 50 (43.5) |
| Diabetes | 34 (29.6) |
| Cataract type | |
| Brunescent | 71 (61.7) |
| White | 44 (38.3) |
| Right eye | 64 (55.7) |
| Image quality of OCT | 0.4 ± 0.9 |
| Axial length, mm | 24.02 ± 1.60 |

Data are expressed as mean ± SD or number (%) of cases.

OCT, optical coherence tomography.

respectively; and 1 (0.9%) with branch retinal vein occlusion, geographic atrophy, peripheral chorioretinal atrophy, retinal detachment, and retinal pigment epithelium tear, respectively (Table 2). Among them, 27 (73%) eyes had the same pathology in the fellow eye; 7 with drusen, 5 with diabetic retinopathy, 4 with myopic degeneration, 3 with glaucoma, 2 with lattice degeneration and exudative age-related macular degeneration, respectively, 1 with epiretinal membrane, branch retinal vein occlusion, peripheral chorioretinal atrophy, and retinal tear, respectively.

Univariate analysis showed that brunescent cataract and pathology in the fellow eye were significantly associated with intraocular pathology in mature cataract eyes ($P = 0.037$ and $P < 0.001$, respectively). Multivariate analysis revealed that pathology in the fellow eye was the only risk factor for intraocular pathology in mature cataract eyes (odds ratio, 47.72; 95% confidence interval, 13.03–174.80; $P < 0.001$) (Table 3).

**Table 2. Prevalence of intraocular pathologies observed after mature cataract surgery.**

| Intraocular pathology | Number of eyes (%) |
|---|---|
| Drusen | 7 (6.1) |
| Myopic degeneration | 6 (5.2) |
| Diabetic retinopathy | 5 (4.3) |
| Glaucoma | 4 (3.5) |
| Epiretinal membrane | 3 (2.6) |
| Lattice degeneration | 3 (2.6) |
| Exudative AMD | 2 (1.7) |
| Retinal tear | 2 (1.7) |
| Branch retinal vein occlusion | 1 (0.9) |
| Geographic atrophy | 1 (0.9) |
| Peripheral chorioretinal atrophy | 1 (0.9) |
| Retinal detachment | 1 (0.9) |
| RPE scar | 1 (0.9) |

AMD, age-related macular degeneration; RPE = retinal pigment epithelium.

**Table 3. Logistic regression analysis for factors associated with intraocular pathologies in eyes with mature cataract.**

| Variable | Univariate analyses | | Multivariate analyses | |
|---|---|---|---|---|
| | OR (95% CI) | *P* | OR (95% CI) | *P* |
| Age (year) | 1.03 (0.99–1.06) | 0.112 | 1.06 (1.00–1.12) | 0.068 |
| Sex, female | 1.22 (0.55–2.72) | 0.631 | | |
| Cataract type, brunescent | 2.53 (1.06–6.07) | *0.037* | 0.87 (0.21–3.06) | 0.846 |
| Axial length | 1.24 (0.97–1.58) | 0.083 | 1.28 (0.87–1.89) | 0.206 |
| Diabetes | 1.22 (0.52–2.85) | 0.643 | | |
| Hypertension | 1.60 (0.73–3.52) | 0.242 | | |
| Fellow eye pathology | 39.42 (12.35–125.84) | *<0.001* | 47.72 (13.03–174.80) | *<0.001* |

Statistically significant values are represented in italics.

Factors with a *P*-value < 0.20 in univariate logistic regression analysis was included in multivariate analysis.

CI, confidence interval; OR, odds ratio.

During the mean follow-up period of 126 days, 5 cases of elevated intraocular pressure and 1 case of cystoid macular edema were noted postoperatively. Otherwise, no cases with epiretinal membrane, endophthalmitis, or retinal detachment were found.

## Discussion

In this study, intraocular pathologies were observed in 37 eyes (32.2%) after mature cataract surgery. The most frequently observed abnormalities were drusen (n = 7, 6.1%), myopic degeneration (n = 6, 5.2%), and diabetic retinopathy (n = 5, 4.3%). Pathology in the fellow eye was the only risk factor for intraocular pathology in eyes with mature cataracts.

The rate of posterior segment pathology on ocular ultrasonography in eyes with dense cataract reported in previous studies ranges from 5.2–30.1% [3–7]. Anteby et al. reported posterior segment pathology on ocular ultrasonography in eyes with dense cataract in 100 (19.6%) out of 509 patients; Salman et al., 36 (8.6%) among 418 eyes; Qureshi et al., 90 (12%) among 750 patients; Bello et al., 6 (5.2%) among 116 eyes; and Mendes et al., 87 (30.1%) among 289 eyes. These reports have indicated limited posterior segment pathologies, including retinal detachment, post staphyloma, and vitreous hemorrhage, using ultrasonography. Our study showed a higher detection rate than that of previous studies because postoperative direct fundus examination is more accurate and detailed than ultrasound. Since the detection rate of posterior segment abnormalities on ultrasonography is higher in cases of ocular trauma and some of the previous studies included ocular trauma patients [6], the gap in the detection rate with ours is considered to be greater. However, this difference could not diminish the clinical implication of preoperative evaluation using ultrasonography in eyes with mature cataract. This is because, even though the frequency is small, it is possible to detect diseases with great clinical significance, such as one case of retinal detachment found in this study, using ultrasonography. Meanwhile, the prevalence of each posterior segment pathology in our study was unremarkable compared with the overall prevalence of the same pathology among similar age groups in South Korea [12–15]. This suggests that the mature cataract itself is an independent factor of intraocular pathologies described in this study. However, although this was not addressed in this study, the detection of posterior segment pathologies could be delayed due to dense cataracts.

In this study, 27 (73%) of the 37 eyes had the same abnormalities in the fellow eye. This seems to be because most posterior segment pathologies identified in this study usually appear in both eyes [16–18].

This study has some limitations. First, because this was a retrospective study that included a small number of patients with mature cataracts in a single center, the possibility of a selection bias exists. Second, since posterior segment pathologies with different risk factors were integrated into one dependent variable (intraocular pathology) in the logistic analysis, caution is required when interpreting the results. Further studies with a larger number of patients are warranted to elucidate the association between mature cataracts and each posterior segment pathology.

In conclusion, the prevalence of postoperative intraocular pathologies was not particularly high in eyes with mature cataracts. Pathology in the fellow eye is indicative of a high risk of intraocular pathology in eyes with mature cataracts. Our results would help patients with mature cataract understand the risk of intraocular pathologies with obvious evidence.

## Supporting information

**S1 Fig. Representative fundus photography and OCT image with brunescent cataract.** A. Wide fundus photography showing indistinguishable retinal structures. B. Only the retinal pigment epithelium layer is barely observed in the OCT image with an image quality of 5. OCT, optical coherence tomography.
(TIF)

**S1 File.**
(XLSX)

## Acknowledgments

The authors alone are responsible for the content and writing of the paper.

## Author Contributions

**Conceptualization:** Kwan Hyuk Cho.

**Data curation:** Min Seok Kim, Jung Hyoo Moon, Kwan Hyuk Cho.

**Formal analysis:** Min Seok Kim, Myung Won Lee.

**Investigation:** Min Seok Kim, Kwan Hyuk Cho.

**Methodology:** Min Seok Kim, Kwan Hyuk Cho.

**Resources:** Min Seok Kim, Jung Hyoo Moon, Myung Won Lee.

**Supervision:** Jung Hyoo Moon, Myung Won Lee, Kwan Hyuk Cho.

**Writing – original draft:** Min Seok Kim.

**Writing – review & editing:** Min Seok Kim, Jung Hyoo Moon, Kwan Hyuk Cho.

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
