## [Decision Letter · Decision Letter 0]

18 Nov 2021

PONE-D-21-32917Analysis of postoperative intraocular pathologies in patients with mature cataractsPLOS ONE

Dear Dr. Cho,

Thank you for submitting your manuscript to PLOS ONE. After careful consideration, we feel that it has merit but does not fully meet PLOS ONE’s publication criteria as it currently stands. Therefore, we invite you to submit a revised version of the manuscript that addresses the points raised during the review process.

We look forward to receiving your revised manuscript.

Kind regards,

Michael Mimouni

Academic Editor

PLOS ONE

Journal Requirements:

Reviewers' comments:

Reviewer's Responses to Questions

**Comments to the Author**

1. Is the manuscript technically sound, and do the data support the conclusions?

Reviewer #1: Yes

Reviewer #2: Yes

2. Has the statistical analysis been performed appropriately and rigorously? 

Reviewer #1: Yes

Reviewer #2: Yes

3. Have the authors made all data underlying the findings in their manuscript fully available?

Reviewer #1: Yes

Reviewer #2: No

4. Is the manuscript presented in an intelligible fashion and written in standard English?

Reviewer #1: Yes

Reviewer #2: Yes

5. Review Comments to the Author

Reviewer #1: Well written and simple article. Conclusions are not surprising. I wonder if the cataract surgery itself could have affected the post op findings (ERM, or retinal tears?). In any event I do not suppose this to carry clinical significance.

Reviewer #2: This is an interesting paper regarding the prevalence and risk factors of intraocular pathologies after mature cataract surgery. I have few questions for the authors:

1.- Although you state in your discussion that limited posterior segment pathologies can be seen on ocular ultrasonography, don’t you think that it is an important study to perform in all patients with dense cataract in which the fundus examination is not possible?

You report that one eye had a retinal detachment (RD) after cataract surgery, in this case, it would be important to have a previous ocular ultrasound, because you do not know if the RD was a postoperative complication of the surgery or it was already present before the surgery.

2.- You report that 73% of the eyes had the same pathology in the fellow eye, could you please specify which pathologies were found binocularly?

6. PLOS authors have the option to publish the peer review history of their article (what does this mean?). If published, this will include your full peer review and any attached files.

Reviewer #1: **Yes: **Victor Flores

Reviewer #2: No

---

## [Author Response · Author response to Decision Letter 0]

26 Nov 2021

Journal Requirements:

Answer: We thank you for the opportunity to submit a revised version of the manuscript. As per your comments, we have checked that our manuscript meets the journal’s style requirements in the pdf files above.

Answer: We have checked that the reference list is complete and correct. 

Answer: We contacted the IRB center again and confirmed that it is possible to upload data without personally identifiable information. We uploaded the data file as ‘supporting information’. Also, we changed the data availability statement as below,

‘All relevant data are within the manuscript and its Supporting information files.’

Answer: The ORCID iD of the corresponding author has been linked. Thank you.

Reviewers' comments:

Reviewer #1: Well written and simple article. Conclusions are not surprising. I wonder if the cataract surgery itself could have affected the post op findings (ERM, or retinal tears?). In any event I do not suppose this to carry clinical significance.

Answer: Thank you for your thoughtful comments. As suggested, we reviewed the chart of all patients again to find any post op findings. During the mean follow-up period of 126 days, 5 cases of elevated intraocular pressure and 1 case of cystoid macular edema were noted postoperatively. Otherwise, no cases with epiretinal membrane, endophthalmitis, or retinal detachment were found. 

As the reviewer commented, we also hypothesized that the prevalence of intraocular pathologies would not be particularly high compared to that of normal population. However, we thought our findings are significant because this is the first report to quantify the frequency and odds ratio of the intraocular pathologies observed after mature cataract surgery. Although our study may not carry great clinical significance, it will be helpful to explain the risks of intraocular pathologies in detail by presenting evidence in this study rather than vague warnings for patients with mature cataracts. We added this point in the result and conclusion section.

In page 9, line 109.

During the mean follow-up period of 126 days, 5 cases of elevated intraocular pressure and 1 case of cystoid macular edema were noted postoperatively. Otherwise, no cases with epiretinal membrane, endophthalmitis, or retinal detachment were found.

In page 11, line 151.

Our results would help patients with mature cataract understand the risk of intraocular pathologies with obvious evidence.

Reviewer #2: This is an interesting paper regarding the prevalence and risk factors of intraocular pathologies after mature cataract surgery. I have few questions for the authors:

1.- Although you state in your discussion that limited posterior segment pathologies can be seen on ocular ultrasonography, don’t you think that it is an important study to perform in all patients with dense cataract in which the fundus examination is not possible?

You report that one eye had a retinal detachment (RD) after cataract surgery, in this case, it would be important to have a previous ocular ultrasound, because you do not know if the RD was a postoperative complication of the surgery or it was already present before the surgery.

Answer: We totally agree with you that ocular ultrasonography still has an important meaning for preoperative evaluation in eyes with mature cataract. As you commented, we added this point in the discussion section.

In page 10, line 128,

However, this difference could not diminish the clinical implication of preoperative evaluation using ultrasonography in eyes with mature cataract. This is because, even though the frequency is small, it is possible to detect diseases with great clinical significance, such as one case of retinal detachment found in this study, using ultrasonography.

2.- You report that 73% of the eyes had the same pathology in the fellow eye, could you please specify which pathologies were found binocularly?

Answer: We apologize for insufficient explanation regarding the results. As your suggestion, we mentioned specific results about eyes that had the same pathology in the fellow eye.

In page 7, line 89,

7 with drusen, 5 with diabetic retinopathy, 4 with myopic degeneration, 3 with glaucoma, 2 with lattice degeneration and exudative age-related macular degeneration, respectively, 1 with epiretinal membrane, branch retinal vein occlusion, peripheral chorioretinal atrophy, and retinal tear, respectively.

Answer: Thank you. We followed your protocol.

---

## [Decision Letter · Decision Letter 1]

17 Jan 2022

Analysis of postoperative intraocular pathologies in patients with mature cataracts

PONE-D-21-32917R1

Dear Dr. Cho,

We’re pleased to inform you that your manuscript has been judged scientifically suitable for publication and will be formally accepted for publication once it meets all outstanding technical requirements.

Kind regards,

Michael Mimouni

Academic Editor

PLOS ONE

Additional Editor Comments (optional):

Reviewers' comments:

Reviewer's Responses to Questions

**Comments to the Author**

1. If the authors have adequately addressed your comments raised in a previous round of review and you feel that this manuscript is now acceptable for publication, you may indicate that here to bypass the “Comments to the Author” section, enter your conflict of interest statement in the “Confidential to Editor” section, and submit your "Accept" recommendation.

Reviewer #1: All comments have been addressed

Reviewer #2: All comments have been addressed

2. Is the manuscript technically sound, and do the data support the conclusions?

Reviewer #1: Yes

Reviewer #2: Yes

3. Has the statistical analysis been performed appropriately and rigorously? 

Reviewer #1: I Don't Know

Reviewer #2: Yes

4. Have the authors made all data underlying the findings in their manuscript fully available?

Reviewer #1: Yes

Reviewer #2: Yes

5. Is the manuscript presented in an intelligible fashion and written in standard English?

Reviewer #1: Yes

Reviewer #2: Yes

6. Review Comments to the Author

Reviewer #1: Thank you for addressing the concerns in the revision. Look forward to seeing your paper published soon.

Reviewer #2: Thank you for addressing my questions and adding that paragraph regarding the ultrasongraphy, as it is an important preoperative exam in patients with dense cataracts.

7. PLOS authors have the option to publish the peer review history of their article (what does this mean?). If published, this will include your full peer review and any attached files.

Reviewer #1: No

Reviewer #2: No

---

## [Editor Report · Acceptance letter]

21 Jan 2022

PONE-D-21-32917R1 

Analysis of postoperative intraocular pathologies in patients with mature cataracts 

Dear Dr. Cho:

I'm pleased to inform you that your manuscript has been deemed suitable for publication in PLOS ONE. Congratulations! Your manuscript is now with our production department. 

Kind regards, 

on behalf of

Dr. Michael Mimouni 

Academic Editor

PLOS ONE